The modulatory effect of Moringa oleifera leaf extract on endogenous antioxidant systems and inflammatory markers in an acetaminophen-induced nephrotoxic mice model

Karthivashan Govindarajan 1
Kura Aminu Umar 1
Arulselvan Palanisamy 1
Md. Isa Norhaszalina 1
Fakurazi Sharida sharida.fakurazi@gmail.com 1 2
1 Institute of Bioscience, Universiti Putra Malaysia , Serdang , Selangor , Malaysia
2 Faculty of Medicine and Health Sciences, Universiti Putra Malaysia , Serdang , Malaysia
Nogueira Cristina
Electronic publication date: 2016 Jul 7
Publication date: 2016
Volume: 4
Electronic Location ID: e2127
Received 2016 Apr 12; Accepted 2016 May 21
Copyright: ©2016 Karthivashan et al.
Copyright year: 2016
Copyright holder: Karthivashan et al.
License: This is an open access article distributed under the terms of the Creative Commons Attribution License, which permits unrestricted use, distribution, reproduction and adaptation in any medium and for any purpose provided that it is properly attributed. For attribution, the original author(s), title, publication source (PeerJ) and either DOI or URL of the article must be cited.
License URL: https://creativecommons.org/licenses/by/4.0/

Keywords: Moringa oleifera, Acetaminophen nephrotoxicity, Serum biochemical markers, Antioxidant enzymes, Inflammatory cytokines, Kidney histology

Funding: Universiti Putra Malaysia GP-IPS/2013/9397300 This research work was supported by research grant from Universiti Putra Malaysia, Project number GP-IPS/2013/9397300. The funders had no role in study design, data collection and analysis, decision to publish, or preparation of the manuscript.

==============================
N-Acetyl-p-Aminophenol (APAP), also known as acetaminophen, is the most commonly used over-the counter analgesic and antipyretic medication. However, its overdose leads to both liver and kidney damage. APAP-induced toxicity is considered as one of the primary causes of acute liver failure; numerous scientific reports have focused majorly on APAP hepatotoxicity. Alternatively, not many works approach APAP nephrotoxicity focusing on both its mechanisms of action and therapeutic exploration. Moringa oleifera (MO) is pervasive in nature, is reported to possess a surplus amount of nutrients, and is enriched with several bioactive candidates including trace elements that act as curatives for various clinical conditions. In this study, we evaluated the nephro-protective potential of MO leaf extract against APAP nephrotoxicity in male Balb/c mice. A single-dose acute oral toxicity design was implemented in this study. Group 2, 3, 4 and 5 received a toxic dose of APAP (400 mg/kg of bw, i.p) and after an hour, these groups were administered with saline (10 mL/kg), silymarin—positive control (100 mg/kg of bw, i.p), MO leaf extract (100 mg/kg of bw, i.p), and MO leaf extract (200 mg/kg bw, i.p) respectively. Group 1 was administered saline (10 mL/kg) during both the sessions. APAP-treated mice exhibited a significant elevation of serum creatinine, blood urea nitrogen, sodium, potassium and chloride levels. A remarkable depletion of antioxidant enzymes such as SOD, CAT and GSH-Px with elevated MDA levels has been observed in APAP treated kidney tissues. They also exhibited a significant rise in pro-inflammatory cytokines (TNF-α, IL-1β, IL-6) and decreased anti-inflammatory (IL-10) cytokine level in the kidney tissues. Disorganized glomerulus and dilated tubules with inflammatory cell infiltration were clearly observed in the histology of APAP treated mice kidneys. All these pathological changes were reversed in a dose-dependent manner after MO leaf extract treatment. Therefore, MO leaf extract has demonstrated some therapeutic effectiveness against APAP-induced nephrotoxicity through enhancement of the endogenous antioxidant system and a modulatory effect on specific inflammatory cytokines in kidney tissues.

Introduction

Acetaminophen (APAP) is a readily available over-the-counter medication used as an effective painkiller and fever suppressor. APAP retains a virtuous safety profile at therapeutic doses. However, when its therapeutic index is breached, it results in acute/chronic hepato-renal damage in both human and experimental animals (Ghosh et al., 2010; Karthivashan, Arulselvan & Fakurazi, 2015; Karthivashan et al., 2015). Though the incident rate of APAP hepatotoxicity is higher than the renal toxicity, the latter leads to 1–2% of acute renal failure in patients with APAP overdose and can be fatal (Eguia & Materson, 1997). The pathophysiology of APAP-induced nephrotoxicity is not much explored compared to APAP hepatotoxicity. Based on previous literature they both allegedly expressed a similar kind of pathophysiology, yet some subtle differences were observed, and remain indistinct (Li et al., 2003; Cekmen et al., 2009; Aycan et al., 2015).

The most probable mechanism of APAP nephrotoxicity involves the metabolic activation of the reactive toxic metabolite, N-acetyl-p-benzoquinone imine (NAPQI). At therapeutic doses, only a few percent of APAP gets converted to the reactive toxic metabolite NAPQI, which is further reduced by glutathione and subsequently excreted as glucuronidated and sulfated (non-toxic) hydrophilic metabolites through the renal system. In an APAP overdose, the supply of sulfate and glutathione get exhausted, thus more NAPQI is generated via CYP450 metabolism. This electrophilic intermediary binds with available cellular proteins and initiate lipid peroxidation, mediated reactive oxygen species (ROS) and other free radical formation, thereby inducing oxidative stress and inflicting renal tissue damage (Isik et al., 2006; Ahmad et al., 2012). This cascade furthermore provokes inflammatory signals and extended the injury, resulting in tubular cell-death / acute renal failure (Möller-Hartmann & Siegers, 1991). Due to its fatal nature, the requirement of an antidote/therapeutic agent against APAP renal toxicity becomes crucial. N-acetylcysteine (NAC), a precursor of GSH, is well known for its hepato-protective nature against APAP-induced hepatotoxicity in both humans and animals; however, it has a limited function towards APAP-induced renal toxicity (Eguia & Materson, 1997; Mazer & Perrone, 2008). Thus, the hunt for alternative, safe and therapeutically effective compounds against APAP-induced renal toxicity is essential.

Moringa oleifera Lam (MO) is a wide-spread tropical and subtropical species belongs to Moringaceae family. It is well known for its remarkable nutritional value and elite therapeutic potential against extensive clinical conditions. Moringa oleifera Lam is commonly known as “drumstick tree” or “horseradish tree” and almost all parts of this plant, including the root, bark, stem, leaves, flowers and pods possess huge amounts of micro- and macronutrients. It provides both animal and human nutritional supplements (Siddhuraju & Becker, 2003; Anwar et al., 2007). It possesses a rich and rare combination of therapeutically-active candidates such as kaempferol, rhamnetin, quercitin, chlorogenic acid, rutin, and apigenin, and is also enriched with an exogenous supply of ascorbic acid and carotenoids, which are renowned antioxidant candidates (Anwar et al., 2007; Karthivashan et al., 2013). MO has been utilized for ages as traditional medicine in the treatment of numerous disorders as an antiseptic, anti-diabetic, antiepileptic, antiparalytic, antiviral, anti-inflammatory effect. Additionally, numerous scientific reports on various parts of the plant have reported on its medicinal value, among which its leaves has been extensively studied in a wide variety of clinical conditions for antimicrobial, anti-inflammatory, anti-cancer, and anti-diabetic effects (Anwar et al., 2007). Our research team has recently identified that flavonoids such as kaempferol, quercitin and apigenin were likely involved in the enhanced antioxidant effect of MO leaves extract, and further established its hepatoprotective mechanism of action against APAP-induced hepatotoxicity (Karthivashan et al., 2015; Karthivashan et al., 2013).

In recent years, it has been established that the existence of trace elements in MO leaf extract also contributes to improvising human health and combating various health disorders (Gowrishankar et al., 2010; Prashanth et al., 2015). Thus, in this study, we evaluated several essential/non-essential trace elements of MO leaf extract to investigate their possible involvement against APAP toxicity. The pathophysiology of APAP-induced hepatotoxicity is proposed to be similar to that of APAP nephrotoxicity; thus, here we extended our investigation on the potential nephro-protective mechanism of MO leaf extract against APAP-induced nephrotoxicity. Furthermore, silymarin has been selected as the positive control for this study, based on previous study reports due to its enhanced hepato- and renal-protective properties against APAP toxicity in mice due to its enriched antioxidative and anti-inflammatory nature (He, Kim & Sharma, 2004; Bektur et al., 2016). This would pave the way for further investigation on the advancement of MO leaf extract as an effective therapy for both APAP-induced nephro- and hepato-toxicity in the field of clinical/translational medication.

Materials and Methods

Chemicals

Acetaminophen and silymarin were procured from Sigma (St. Louis, MO, USA). All kidney function markers kits, malondialdehyde (MDA) and antioxidant enzyme assay kits were purchased from Roche Diagnostics (Germany), Biovision Research kits (Milpitas, CA, USA) and Cayman Chemical Company (Cayman Chemical, Ann Arbor, MI, USA) respectively. HEPES buffer was obtained from Nacalai Tesque (Kyoto, Japan). Porcelain crucible, analytical balance (OHAUS, made in Switzerland), oven (Genlab, Cheshire, UK), type 1500 furnace, desiccators, and Solaar M atomic absorption spectrometer (AAS) (Thermo Elemental, Waltham, MA, USA) were used for AAS analysis. All glassware used was rinsed and soaked in 10% (v/v) HNO3 overnight. They were rinsed with de-ionized water and dried before use. All other chemicals and reagents used were obtained from Sigma (St. Louis, MO, USA) unless indicated otherwise.

Plant materials

Fresh mature leaves from the Moringa oleifera tree were harvested from Garden-2, Universiti Putra Malaysia (UPM) and have been confirmed similar to the voucher specimen (SK 1561/08) previously deposited in the Institute of Bioscience, UPM (IBS) Herbarium unit. The whole plant leaves were collected, washed in running tap water, air dried at room temperature (24 °C) for a day and oven dried for two consecutive days at 45 °C. The dried plant material was ground using a mechanical blender and stored in an airtight container after processing.

Preparation of leaf extract

The Moringa oleifera leaf powder was macerated exhaustively with 90% ethanol (ethanol: distilled water, 90:10) in aspirator bottle for three consecutive days at room temperature with continuous shaking. The residue was strained and the filtrate was condensed using a rotary evaporator at 40 °C. The condensed residue was of slurry nature and dark green in color, which were further freeze-dried. The obtained freeze-dried extracts were weighed, kept in a capped container, labeled appropriately and stored at −20 °C.

Preliminary analysis of trace elements

Sample digestion—dry ashing method

One gram of MO leaf extract was placed in a porcelain crucible in a furnace. The furnace temperature was steadily increased from room temperature to 350 °C. The sample turned to ash after 4 h and the process continued until whitish grey ash residue was attained. The residue was dissolved in 5 ml of nitric oxide and increased to 10 mL volume in appropriate volumetric flask.

Determination of trace elements

In this study, we evaluated three essential trace elements; Copper (Cu), Manganese (Mn), Nickel (Ni), and three toxic trace elements; Cromium (Cr), Lead (Pb) and Cadmium (Cd) present in the MO leaf extract. Working standard solutions of appropriate elements were prepared from stock standard solution (1,000 mg/L) and absorbance values were obtained for various working standards for each element in the samples, using an atomic absorption spectrometer (AAS). The absorbance values were plotted against concentration, whereby the formed linear calibration curves revealed the actual concentration of the sample. A blank reading was also taken and essential correction was made during the calculation of concentration of various elements.

Animals and experimental design

Male Balb/c mice of 25–30 g weight (10–12 weeks old) were handled at the Animal House Unit, Faculty of Medicine, Universiti Putra Malaysia (UPM). Animals were acclimatized for a week at 26 ± 2 °C with a 12 h light/dark cycle. Free access to food and water was allowed at all times. During the acclimatization period, five mice were housed per cage in plastic cages using homogenized wood shavings as bedding. All experimental protocols used on the animals were done with the approval (UPM/IACUC/AUP-17/2013) and standard ethical guidelines of the Faculty of Medicine and Health Sciences, Universiti Putra Malaysia, Malaysia, IACUC (Institutional Animal Care and Use Committee) were followed.

A single-dose acute oral toxicity design was performed on Balb/c mice in this study. On Day 8, following acclimatization, the mice were randomly assigned into five groups (n = 6 per group) with same housing setup. The APAP was dissolved in an appropriate concentration in warm saline. Group 2, 3, 4 and 5 received a toxic dose of APAP (400 mg/kg of bw, i.p) followed by administration of saline (10 mL/kg), silymarin (100 mg/kg of bw, i.p), MO leaf extract (100 mg/kg of bw, i.p) and MO leaf extract (200 mg/kg bw, i.p) respectively, an hour after the administration of APAP lethal dose. Group 1 was administered saline (10 mL/kg) during the two sessions. The animals were sacrificed using diethyl ether about 24 h after induction of APAP toxicity and subsequent treatment with the respective doses of silymarin/MO extract. Blood samples were rapidly obtained by cardiac puncture, and serum was prepared and stored at −20 °C. The kidney was collected, snap-frozen and stored immediately at −80 °C and a portion of them was fixed at 10% buffered formalin.

Biochemical parameters

All biochemical assays were performed spectrophotometrically using a Hitachi-912 Autoanalyser (Mannheim, Germany) with kits supplied by Roche Diagnostics (Mannheim, Germany). Indicators of kidney function, including serum creatinine, urea, sodium (Na+), potassium (K+), and chloride (Cl−) levels were measured. In order to obtain data with good sensitivity and validity, serum samples were analyzed blindly and in triplicate.

Histopathological examination

Renal tissues from each group were fixed in 10% formalin and fixed samples were embedded in paraffin, sectioned in 5 µm-thick sections and stained with hematoxylin-eosin (H&E) stain. All the pathological changes in renal tissues were examined and photographed using an Olympus microscope (BX-51; Olympus, Tokyo, Japan).

Measurement of kidney oxidative stress and inflammatory markers

The frozen renal tissue was thawed and homogenized in 10% (w/v) with ice-cold 0.1 M phosphate buffer saline and centrifuged at 9,000 rpm for 20 min at 4 °C, and the supernatants were assayed according to the instructions provided by the manufacturer. BioVision Research kits (Milpitas, CA, USA) were used to determine the MDA level, and the activity of anti-oxidant enzymes were determined using SOD, CAT and GPx kits obtained from the Cayman Chemical Company (Ann Arbor, MI, USA). The levels of pro-inflammatory (TNF-α, IL-1β, IL-6) and anti-inflammatory (IL-10) cytokines in renal tissue homogenates were determined using commercially available ELISA kits, in accordance with manufacturers’ instructions (R&D, Mannheim, Germany).

Statistical analysis

The results are expressed as mean ± SEM. The normal distribution of the data was confirmed by Shapiro–Wilk test using GraphPad software 5.0 (GraphPad, La Jolla, CA). Statistical analyses, such as one-way ANOVAs and associated Student t-tests were performed for the biochemical, oxidative stress and inflammatory parameters using Excel software (Microsoft, Redmond, WA, USA). A p-value less than 0.05 considered as statistically significant.

Results

Trace elements of MO leaf extract

The MO leaf extract were found to contain various trace elements, which aid numerous biochemical processes in the human body. In the present study, six trace elements, namely Cu, Mn, Ni, Cr, Pb and Cd were determined with substantial accuracy. The concentration of each element was determined, and the corresponding linear calibration curves were obtained and are reported in Table 1, with their biochemical functions. The dried MO leaf extract contain high amount of Manganese (Mn) and Copper (Cu) with the value of 36.157 ± 0.037 and 12.323 ± 0.098 mg/kg of dry leaf extract, respectively compare to Nickel (Ni) with the value of 1.657 ± 0.008 mg/kg of dry leaf extract. Toxic trace elements such as Chromium (Cr), Cadmium (Cd) and Lead (Pb) were expressed in a negligible amount, which was found to be less than 0.005 mg/kg of dry leaf extract.

Table 1 Selective trace elemental composition of dried Moringa (M. oleifera Lam.) leaves.

Trace elements	Concentration (mg/kg of dry leaf extract)	Function	
Copper	12.323 ± 0.098	A catalytic cofactor in the redox chemistry of free radical scavenging	
Manganese	36.157 ± 0.037	Activator of several manganese metalloenzymes and one form of antioxidant enzyme superoxide dismutase (SOD).	
Nickel	1.657 ± 0.008	Aids in iron absorption, as well as adrenaline and glucose metabolism, hormones, lipid, cell membrane and improves bone strength	
Chromium (VI)	<0.005 ± 0.004	Causes gastrointestinal effects in humans and animals, including abdominal pain, vomiting, and hemorrhage.	
Lead	<0.005 ± 0.002	Lead has no known preferred function in the body, but accumulation of lead is highly toxic for human body.	
Cadmium	<0.005 ± 0.005	Cadmium is extremely toxic. It mainly affects the kidney, the cardiovascular system, and is related to cancer.	

MO leaf extract minimizes APAP-induced nephrotoxicity in mice

The serum obtained from mice treated with toxic dose of APAP revealed significant (p < 0.05) elevation in creatinine (0.51 ± 0.02 mg/dL), blood urea nitrogen (42.0 ± 2.0 mg/dL), sodium (Na+; 142.20 ± 0.35 mEq/L) potassium (K+; 12.30 ± 0.24 mEq/L), and chloride (Cl−; 106.80 ± 0.84 mEq/L) levels, compare to the control group which displayed the following values: creatinine (0.25 ± 0.01 mg/dL), blood urea nitrogen (19.60 ± 1.4 mg/dL), sodium (Na+; 138.10 ± 0.45 mEq/L) potassium (K+; 11.10 ± 0.16 mEq/L) and chloride (Cl−; 103.70 ± 1.08 mEq/L). Mice treated with MO extract had lowered levels of serum creatinine, blood urea nitrogen, Na+, K+, and Cl− compared to the groups that were not treated with MO, and the reduction was found to be dose-dependent. Among the adapted two doses of MO leaf extract, mice treated with 200 mg/kg of bw displayed significant (p < 0.05) decrease in serum kidney biomarkers with the values of creatinine (0.29 ± 0.04 mg/dL), blood urea nitrogen (30.8 ± 1.0 mg/dL), sodium (Na+; 140.20 ± 0.18 mEq/L), potassium (K+; 11.30 ± 0.12 mEq/L and chloride (Cl−; 104.0 ± 0.42 mEq/L and it proximate the effects of the silymarin treated (positive control) group (Figs. 1A–1E).

Figure 1 Modulation of Serum biochemical markers and electrolytes level.

Moringa oleifera leaves suppress the detrimental effect of APAP induced nephrotoxicity. (A–E) represents the level of serum biochemical markers such as creatinine, urea and electolytes such as sodium, potassium and chloride. Values are expressed as the mean ± SEM of n = 6 mice in each group. The normality distribution of the data has been confirmed by Shapiro-Wilk test and statistical analysis was performed using one-way ANOVA associated student t-test. *P < 0.05 compared to the control group, $P < 0.05 compared to APAP administered group. Among the treatment groups, when the protection is total (i.e., different from acetaminophen group and relevant to control group) or partial (i.e., different from acetaminophen and control groups) were represented as # or ¥ respectively.

The histological micrographs of APAP intoxicated mice kidney sections portrayed further, renal tissue damage by exhibiting severely disorganized glomerulus, dilated tubules, presence of granular casts and inflammatory cell infiltrates (Fig. 4C). Histological analysis of MO (100 mg/kg bw) treated mice kidney sections showed some sparsely disorganized glomerulus, tubular dilation with moderate tubular casting and inflammation (Fig. 4E). However at a higher dose of 200 mg/kg of MO (Fig. 4G), the glomerular and tubular architecture were well preserved. They showed negligible amount of granular casting in the renal tubules, similar to that observed in the positive control (Fig. 4I) group and closely resembles the untreated sham group (Fig. 4A). Subsequently, scores were awarded to the histology images. Compared to the control group, a substantial elevation in the scores was observed in APAP-treated group (negative control). In contrast, MO (100 mg/kg bw) treated mice scored lower than the APAP-treated group. At 200 mg/kg bw, MO treated group equaled the score of positive control group (Fig. 4K).

Figure 2 Level of antioxidant enzymes in kidney tissue.

Dose dependent effect of Moringa oleifera (MO) leaves extract against APAP intoxicated mice kidney via augmentation of endogenous antioxidant status: (A) lipid peroxidation activity (MDA); endogenous antioxidant enzyme levels ((B)-SOD, (C)-CAT & (D)-GPx). Values are expressed as the mean ± SEM of n = 6 mice in each group. The normality distribution of the data has been confirmed by Shapiro-Wilk test and statistical analysis was performed using one-way ANOVA associated student t-test. *P < 0.05 compared to control group, $P < 0.05 compared to APAP group. Among the treatment groups, when the protection is total (different from acetaminophen group and relevant to control group) or partial (different from acetaminophen and control groups) were represented as # or ¥ respectively.

MO leaf extract regulates and restores the antioxidant status, in APAP-induced nephrotoxic mice

The level of MDA, SOD, CAT and GPx activities in the renal samples are presented in Fig. 2. The renal MDA level of APAP group increased significantly (F = 31.63; p < 0.05) to 1.44 ± 0.17 nmol/mg of tissue compared with the control group, which was 0.76 ± 0.13 nmol/mg of tissue, whereas, the MO (200 mg/kg of bw) and silymarin treated groups showed a significant (F = 30.29, 41.6; p < 0.05) decrease in the level of MDA with the values around 0.80 ± 0.11 nmol/mg of tissue (Fig. 2A). Kidneys obtained from the mice intoxicated with APAP showed significant decrease in the SOD (47.17 ± 5.05 U/mg), CAT (65.83 ± 5.54 nmol/ min/mg of protein) and GPx (2.55 ± 1.12 nmol/min/mg of protein) when compared with control groups showing values of 63.83 ± 6.00 U/mg, 104.27 ± 5.54 and 4.25 ± 0.26 nmol/min/mg of protein respectively (F = 8.26, 96.29, 6.24; p < 0.05). On the contrary, SOD, CAT and GPx activities of the groups treated with MO leaf extract demonstrated a dose-dependent increase, as shown in Figs. 2B–2D. At the higher dose (200 mg/kg of bw), MO leaves extract exhibited values of 93.33 ± 5.01 U/mg, 92.33 ± 9.20 and 7.64 ± 0.33 nmol/ min/mg of protein of SOD, CAT and GPx activities respectively, which were significantly (F = 112.84, 18.73, 12.85; p < 0.05) higher than the APAP-intoxicated mice kidney. The silymarin-(positive control) treated group exhibited SOD, CAT and GPx activity of 63.83 ± 4.01 U/mg, 104.27 ± 3.41 and 6.79 ± 0.16 nmol/min/mg of protein respectively. However, from Figs. 2B and 2D, it was clear that treatment with MO leaf extract greatly exceeded the level of SOD and GPx activity compared to the silymarin-treated group. These results indicate that MO leaf extracts effectively restore the antioxidant status of APAP-intoxicated mice kidney.

MO leaves extract modulates pro/anti-inflammatory cytokines in APAP-induced nephrotoxic mice

To further understand the mechanism of action of MO leaf extract, we evaluated its role in altering the inflammatory cytokines level as displayed in Figs. 3A–3D. We also compared these inflammatory changes with the microscopic evidence of granular cast and inflammatory cell infiltrate into the renal tissues (Figs. 4B–4J). The kidneys of APAP-intoxicated mice showed a significant (F = 24.20, 89.71, 112.95; p < 0.05) rise in the level of the pro inflammatory cytokines TNF-α (416.67 ± 44.93), IL-1β (251.50 ± 34.18), and IL-6 (441.76 ± 19.98) ng/mg of protein, compared to the control group with the values of 132.91 ± 60.16, 28.16 ± 14.24, 219.94 ± 18.68 ng/mg of protein respectively. At the same time, treatment with APAP suppressed the level of anti-inflammatory cytokine IL-10 (131.03 ± 31.81 ng/mg of protein) compared to the control (162.77 ± 22.46 ng/mg of protein). Contrarily, MO leaf extract showed a dose-dependent modulation in the level of these inflammatory markers. Specifically, at 200 mg/kg of bw, MO leaf extract significantly (F = 15.74, 112.73, 15.58; p < 0.05) suppressed the level of TNF-α (292.50 ± 60.71), IL-1β (86.22 ± 21.75), and IL-6 (314.18 ± 52.76) ng/mg of protein, and enhanced the level of IL-10 (215.63 ± 12.34 ng/mg of protein). From Fig. 5, it was clear that the reduction of inflammatory markers in MO treated group exceeded the effects seen in silymarin (positive control) treatment, where the silymarin-treated group exhibited levels of TNF-α (309.58 ± 59.61), IL-1β (102.61 ± 10.13), and IL-6 (306.60 ± 29.82) ng/mg of protein, and the level of anti-inflammatory cytokine IL-10 with a value of 147.69 ± 12.34 ng/mg of protein. This was supported by the histology micrographs (Fig. 4D) of APAP-treated mice, where significant inflammatory cell infiltrate and tubular casting were observed. In the case of MO (200 mg/kg of bw) and silymarin (positive control) treated mice kidneys, meager/negligible inflammatory cell infiltrate was observed.

Figure 3 Alteration in the level of serum inflammatory markers.

Modulatory effect of Moringa oleifera (M.O) leaves extract against APAP intoxicated kidney inflammatory cytokines –(A) TNF-α; (B) IL-1β; (C) IL-6; (D) IL-10. Values are expressed as the mean ± SEM of n = 6 mice in each group. The normality distribution of the data has been confirmed by Shapiro-Wilk test and statistical analysis was performed using one-way ANOVA associated student t-test. Results are shown as the mean ± SEM; *P < 0.05 compared to control group, $P < 0.05 compared to PCM group. Among the treatment groups, when the protection is total (different from acetaminophen group and relevant to control group) or partial (different from acetaminophen and control groups) were represented as # or ¥ respectively.

Figure 4 Photographs of Histopathological modification in kidney tissue.

Photographic sections (H&E 20X –1; 40X –2) of the mice kidney, (A) Control group showing normal histological architecture of tubules (T) and glomerulus (G) (B) Flawless tubules with striated border (T), intact glomerulus (arrow) with surrounding Bowman’s capsule (line). (C) Kidneys of mice treated with APAP showing disorganized glomerulus (DG), dilated tubules (DT) with tubular casting (C) and inflammation (I) (D) severely disorganized glomerulus (dotted arrow), tubular dilation (arrow head), and inflammatory casting were observed (E) Kidney of mice treated with APAP and M.O 100 mg/kg showing sparsely disorganized glomerulus (DG), dilated tubules (DT) with moderate tubular casting (C) and inflammation (I) (F) few disorganized glomerulus (dotted arrow) and tubular dilation (arrow head) with tubular casting (C) and inflammation (I) are noticed. (G) Kidneys of mice treated with APAP and M.O 200 mg/kg showing preserved glomerulus (G) and tubules (T) architecture with mild tubular casting (C) and inflammation (I) (H) mild tubular dilation (arrow head) with tubular casting (C) and inflammation (I) are noticed. (I) Kidneys of mice treated with APAP and silymarin (positive control) showing preserved glomerulus (G) and tubules (T) architecture (J) tubules with striated border (T), intact glomerulus (arrow) are noticed. (K) The histological changes were graded as: (−) score (negative score): no any structural damage, (+) score (one positive score): marginal damage, (+ + ) score (two positive score): moderate damage, (+ + + ) score (three positive score): intense damage.

Figure 5 Potential mechanism of action of MO leaves extract against APAP induced renal toxicity pathway.

Mechanism of action of MO leaves extract against APAP induced renal toxicity pathway: The active constituents and essential trace elements of MO leaves extract successfully enhance the GSH-GPx and endogenous antioxidant system thereby inhibit the oxidative stress mediated renal impairment, induced by APAP overdose. Despite this, MO leaves extract also extensively inhibit the inflammatory cascade by effectual modulation of inflammatory cytokines, thus curbing the further exacerbation of renal injury mediated by inflammatory cytokines. These features evidently project MO leaves extract as a successful nephro-protective agent.

Discussion

Human and animal bodies contain a certain quantity of trace elements, mostly located in the liver, bones and blood. Enzymes like arginase, mitochondrial superoxide dismutase, cholinesterase, phosphoglucomutase, pyruvate carboxylase and several phosphates, peptidases and glycosyltransferases function with aid of these elements as co-factors (Jarapala, Kandlakunta & Thingnganing, 2014). Trace elements are minute in quantity, yet play a vital role in biochemical processes. It has been well established that copper (Cu) and manganese (Mn) are highly known catalytic co-factors for Cu/Zn-SOD and Mn-SOD antioxidant enzymes that enhances the free-radical scavenging activity, thereby ameliorating the effects of oxidative metabolism. Copper is also necessary for both Fe and energy metabolism; it also acts as a reductant in the enzymes lysil oxidase, cytochrome oxidase, dopamine hydroxylase, superoxide dismutase (Harris, 1992; Dichi, Breganó & Cecchini, 2014). In this study, both Cu and Mn were found to be highly present in MO leaf extract compared to the other notable trace elements. However, negligible amounts of toxic elements were found in the MO leaf extract. Nickel (Ni) which has been categorized as a “probable essential trace element” has also been found in the MO leaf extract. Recent reports indicated that Ni functions either as a cofactor facilitating the intestinal absorption of the Fe3+ ion, or alters membrane properties and influences oxidation/reduction systems (Samal & Mishra, 2011; Prashanth et al., 2015). In this study, the elemental analysis results of MO leaves show high amounts of Manganese (Mn) and Copper (Cu) and considerable amounts of nickel (Ni) which might involve assisting the enhancement of endogenous antioxidant system to combat APAP nephrotoxicity. The expression of negligible amount of toxic trace elements in MO leaves suggests its safety aspects in biological systems.

Acetaminophen (APAP), as the most common and established pain relieving and antipyretic medication on the market, disclose safety breach during overdose and results in hepato-renal damage. The incident rate of APAP hepatotoxicity is higher than renal toxicity, however, recent reports highlighted that renal impairment can be lethal (Eguia & Materson, 1997; Ghosh et al., 2010; Karthivashan et al., 2015). During APAP overdose, there is saturation of hepatic metabolic pathways and reduced liver clearance of APAP, allowing for higher amounts of the unmetabolized toxic intermediate, N-acetyl-para-amino-benzoquinoneimine (NAPQI), to come into contact with the kidneys. These toxic metabolites are involved in protein arylation, precisely in the S3 segment of the proximal tubule, thereby initiating renal tubular cell death (Tarloff & Kinter, 1997; Bjorck, Svalander & Aurell, 1988).

Elevations in serum creatinine, blood urea nitrogen, sodium, potassium and chloride levels are the most occurring changes seen in APAP-induced nephrotoxicity (Pradhan et al., 2013; Sebastian, Baskin & Czerwinski, 2007). During renal damage, accumulation of serum urea occurs when the rate of serum urea production exceeds the rate of its clearance; whereas endogenous breakdown of tissue creatine leads to elevation of serum creatinine levels and other electrolytes (Palani et al., 2010). Previous study reports showed that APAP-induced renal damage leads to serum osmolality of Na+, K+, and Cl−, and was significantly increased at 12 h and further increased at 24 h, which was supposedly due to renal hemodynamic compromise and tubular function impairment (Goddard, Strachan & Bateman, 2003; Pakravan et al., 2015). Thus the serum concentration of these parameters serves as the most reliable biomarkers of renal dysfunction. In accordance, the results of this study indicated that administration of APAP-inflicted substantial renal damage as evidenced by the elevated levels of serum creatinine, blood urea nitrogen, Na+, K+, and Cl−. However, mice treated with varying doses of MO leaves extract presented with significantly reduced levels of serum creatinine, blood urea nitrogen, Na+, K+, and Cl−. The values for these were equivalent or even less than the silymarin (positive control) treated group.

This was similar to a previous study report, whereby ethanol extract of Citrus macroptera (EECM) effectively restored the serum biomarkers and electrolytes level, thereby curbing the deterioration caused by alterations of serum Na+, K+, and Cl− levels in APAP-inflicted renal impairment (Paul et al., 2016). The renal histological results correlate well with the alterations noted in biochemical parameters. The APAP-intoxicated mice revealed severely disorganized glomerulus, dilated tubules, and inflammatory casting, which is in agreement with the previous studies (Hamid et al., 2012; Ahmad et al., 2012), whereas MO leaf extract at a higher dose (200 mg/kg) preserved glomerulus and tubular architecture with insignificant tubular casting, similar to the positive control/sham group. However, the histological findings in APAP-intoxicated mice were at variance with that of (Sharifudin et al., 2013), where male Sprague-Dawley rats administered 7 g/kg body weight of APAP did not reveal any significant changes in kidney histology. The observed variance could be attributed to differences in the doses administered, and animal models (Hook, 1993).

To analyze the NAPQI, mediated intracellular reactive oxygen species (ROS) production in the kidney tissues, lipid peroxidation (MDA) levels, and activities of the antioxidant enzymes (SOD, CAT and GPx) were measured. During APAP overdose, an imbalance occurs between the formations of ROS and its scavenging mechanism through the endogenous antioxidant system (Hook, 1993; Ozbek, 2012). This causes an oxidative stress environment, leading to cellular damage via peroxyl radical formation, which is further reorganized through a cyclization process to endoperoxides, and produces malondialdehyde (MDA) as the final product (Yin, Xu & Porter, 2011). In this study the APAP-treated mice showed a significant elevation in MDA levels and substantial decrease in SOD, CAT, and GPx activities, when compared to normal control groups. However, administration of MO leaf extract significantly decreased the levels of MDA and efficiently elevated SOD, CAT, and GPx activities, compared to the APAP-treated group. Our research team has previously reported the existence of flavonoids such as kaempferol, apigenin, quercetin, and multiflorin in the MO leaf extract, which are likely responsible for advancing antioxidant potential.

It was well established that APAP overdose-induced nephrotoxicity occurred via the formation of NAPQI, but some recent reports strongly suggested the role of inflammatory responses in the progression of renal injury (Ghosh et al., 2010; Samal & Mishra, 2011; Ozbek, 2012). APAP intoxication induces oxidative stress mediated renal damage, which further triggers a secondary inflammatory cascade associated with cytokine release from Kupffer cells. Pro-inflammatory cytokines like TNF-α, IL-1β and IL-6 are prominently reported in APAP-induced hepato-renal toxicity. They are engaged in massive tubular infiltration of leukocytes, thereby inducing a sterile inflammatory environment and further exaggerating the renal damage (Hörl, 2010; Sanz et al., 2008). A recent study reported N-acetylcysteine and ozone therapy exhibited effective anti-inflammatory activity based on its inhibitory activities against the expression of TNF-α in APAP-intoxicated mice kidney (Ucar et al., 2013). In another study, bazhen decoction was reported to possess some protective role against APAP toxicity through suppression of various pro-inflammatory cytokines, notably TNF-α, IL-1β and IL-6 (Song et al., 2014). The results of our study showed elevated levels of TNF-α, IL-1β, IL-6 in APAP-administered mice kidney, which was significantly suppressed by the MO leaves extract. Anti-inflammatory cytokines such as IL-10 were also produced by the Kupffer cells at the site of tubular inflammation to check this detrimental influence.

Ucar et al. (2013) reported that n-acetylcysteine and ozone therapy also played a renal-protective role in APAP toxicity by significantly elevating IL-10 cytokine level. Our results consistently indicated significant suppression of IL-10 cytokine in APAP-intoxicated mice, which was effectually restored by MO leaf extract in a dose-dependent manner. Thus, MO leaf extract protects the inflammatory-mediated exacerbation of renal damage in APAP-intoxicated mice by modulation of both pro and anti-inflammatory cytokine level. Thus, the postulated overall mechanism of action of MO leaf extract against APAP-induced nephro-toxicity pathway has been clearly elucidated in this study (Fig. 5).

Conclusion

The MO leaf extracts shielded kidneys from APAP toxicity through enhancement of the endogenous antioxidant system/enzymatic level to counteract the oxidative stress environment (ROS). Certain naturally incorporated bioactive constituents, and highly accessible essential trace elements present in MO leaves aided the renal protective activity. The MO leaf extract also exhibited modulatory effect on specific inflammatory cytokines, and aided in combating the inflammatory cascade associated renal damage seen in APAP toxicity. In the light of biochemical results and histological findings, MO leaf extract can be suggested as a convincing remedy against APAP-induced nephrotoxicity. Thus, further broad translational investigation of these promising protective effects of MO leaves against APAP-induced renal injury may have a substantial influence on developing clinically-feasible strategies to treat patients with renal impairment, or as a supplemental treatment to aid several nephrotoxic drugs on widening of their therapeutic index.

Supplemental Information

Data S1 Raw data compilation

Raw data compilation

Click here for additional data file.

Abbreviations

MO Moringa oleifera

APAP Acetaminophen

GSH-Px Glutathione peroxidase

SOD Superoxide dismutase

CAT Catalase

MDA Malondialdehyde

NAPQI N-acetyl-p-benzoquinoneimine

TNF-α Tumor necrosis factor-α

IL Interleukin

NAC N-acetylcysteine

Additional Information and Declarations

Competing Interests

Author Contributions

Animal Ethics

Data Availability

The authors declare there are no competing interests.

Govindarajan Karthivashan conceived and designed the experiments, performed the experiments, analyzed the data, wrote the paper, prepared figures and/or tables.

Aminu Umar Kura reviewed drafts of the paper.

Palanisamy Arulselvan conceived and designed the experiments, analyzed the data.

Norhaszalina Md. Isa performed the experiments.

Sharida Fakurazi conceived and designed the experiments, analyzed the data, contributed reagents/materials/analysis tools, reviewed drafts of the paper.

The following information was supplied relating to ethical approvals (i.e., approving body and any reference numbers):

All animal experiments were conducted with the approval (UPM/IACUC/AUP-17/2013) and guidelines of the IACUC (Institutional Animal Care and Use Committee) standard ethical guidelines, Faculty of Medicine and Health Sciences, Universiti Putra Malaysia, Malaysia.

The following information was supplied regarding data availability:

The raw data has been supplied as Data S1.

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
