# Peer review of "The modulatory effect of Moringa oleifera leaf extract on endogenous antioxidant systems and inflammatory markers in an acetaminophen-induced nephrotoxic mice model"

_PeerJ, doi:10.7717/peerj.2127_

## Round 0.1 · original submission · Minor Revisions

Please make sure that the paper is read and improved by a native English speaker.

Reviewer 1 ·

Basic reporting

No Comments

Experimental design

No Comments

Validity of the findings

No Comments

Additional comments

This study by Karthivashan et al. demonstrated the protective effect of Moringa oleifera leaves extract on acetaminophen-induced nephro-toxicity in mice. The study is well designed and interesting, and I recommend its publication on PeerJ.

Reviewer 2 ·

Basic reporting

No comments

Experimental design

No comments

Validity of the findings

No comments

Additional comments

This manuscript (#10018) reported the protective potential of Moringa oleifera leaf extract (MO) in a mice model of acetaminophen-induced nephrotoxicity addressing possible therapeutic mechanisms. Renal damage markers as serum urea and creatinine levels as well serum electrolyte levels and histological architecture analysis of kidney were assessed to characterize renal damage. Further, parameters related to oxidative status as MDA levels and antioxidant enzyme activities and to pro/anti-inflammatory condition (e.g. IL) were also assayed to found a mechanistic pathway of MO renoprotection. Moreover, trace elements of biological importance (Mn, Cu, Ni) were found in MO leaf extract.
The findings are interesting and added important information about MO as a potential therapeutic agent. The idea of work is relevant and the manuscript seems well written and structured. However, specific aspects deserve attention. I would like to address some questions and suggestions, which could contribute to the improvement of this paper. I also suggest additional experiments as follows.
 Title
1. The mechanistic idea of this paper should be included in the title (to make it more informative).

 Abstract
1. The expression “Acetaminophen (APAP)” should be changed by “N-Acetyl-p-Aminophenol (APAP), known as acetaminophen…”
2. The authors affirm that “only minimal reports exist on both the mechanism of action and therapeutic exploration for APAP nephrotoxicity”. Indeed, most scientific articles generally focus on aspects linked to liver damage rather than kidney. However, the expression “minimal” does not seem adequate. Authors could use "not many works approach this subject" or other expressions in this regard.
3. Literature information could be reduced to better detail experimental approaches used. For example, animal sex is not reported in the abstract. Treatment schedule with the drugs (dosage, administration route and times of pre-treatment) is not described. Further, it should be indicated that cytokines analysis was performed in kidney tissue (because some parameters were assayed in serum). Positive control employment could also be reported. Important details are missing from this section.
4. Please, include the probable mechanisms of MO renoprotection at conclusion level of abstract.

 Materials and methods
1. Please, include the approval protocol number of Institutional Animal Care and Use Committee.
2. Trace elements of biological importance for enzyme activities as Ni, Mn and Cu were found in MO leaf extract. However, it would be of great value to include zinc (Zn) and selenium (Se) measures to the data set. Indeed, Se research has attracted interest because of its important role in antioxidant selenoproteins (as GPx) for protection against oxidative stress. Zn is also important for SOD function given that it is a cuprozinc enzyme. Therefore, analysis of Zn and Se in MO leaf extract should be included if possible. Of course the ideal approach would to analyze these trace elements in renal tissue (ex vivo) comparing MO and control groups.
3. As reported in a previous study, hepatoprotective action of this plant was investigated by the authors using similar experimental approaches (experimental groups, number of animals, drug dosage/administration route and times). Were the same animals employed for the present study? The division of study brings no problem, but I believe that additional experiments could refine this work.
4. I strongly suggest including GSH dosages and renal Na+K+ATPase activity.
5. Why experimental groups containing MO per se were not evaluated? These groups could be added.
6. Concerning statistical analysis, was the normality of data tested for choosing of statistical test? Its use is important to select between parametric or non-parametric tests. Further, authors indicated the use of t-test or ANOVA (one-way???), but not indicated when each test was used. Please, clarify the information and also include in figure legends. It’s suggested to add the statistical software employed in methods.

 Results
1. I suggest to insert the F values (including degrees of freedom) in the textual description of results
2. Please, indicate when protection is total (different from acetaminophen group and equal to control group) or partial (different from acetaminophen and control groups) through statistical symbols.
3. Please, check the statistical analysis regarding IL10 (control x acetaminophen). Are there significant differences between control and acetaminophen group? Was one-way ANOVA employed?

 Discussion
1. The results about Na, K and Cl levels should be more discussed. They almost go unnoticed in the text.
2. P values and reference to figures (except for figure 5) could be restricted to result section.
3. In figure 5 an increase of GSH is indicated in green (MO extract protective pathway). However, GPx activity (not GSH levels) was assayed. This green indication would be valid only after additional experiments.

Reviewer 3 ·

Basic reporting

No comments.

Experimental design

The present study entitled Acetaminophen induced nephro-toxicity in mice: potential defensive mechanism of Moringa oleífera leaves extract addressed an interesting topic. It shows the therapeutic effect of MO against nephrotoxicity induced by acetaminophen.
The paper of Karthivashan et al. shows the highest quality and rigor necessary to publish in PeerJ. However, I have some questions:
1) There are data about photochemistry profile of MO? The benefic effect of MO extract could be by its photochemistry content?
2) The per se effect of MO did not was tested? It would be interesting to know about its effect alone?

Validity of the findings

No comments.

Additional comments

Minor revisions:
1) Delete “[90%]” in line 143.
2) In line 131 HNO3 should be HNO3. And in line 306 Fe3+ should be Fe3+

---

## Round 0.2 · Minor Revisions

Thank you for submitting the above mentioned manuscript to Peer J. We are pleased to inform you that your paper has been found suitable for publication, providing you can change and amend it according to the enclosed suggestions.

Most of my comments were addressed adequately. However, I have some minor comments that should be addressed before acceptance of the paper.

Although the authors changed the manuscript title to attend the reviewer´s#2 query, I suggest a revision for the title to become concise, informative and tell what the paper is about and what it found.

Moringa oleifera leaves extract modulates antioxidant system and inflammatory cytokines in a model of nephrotoxicity induced by acetaminophen in mice

I advise the authors to find a native English speaker to proofread the manuscript.

---

## Round 0.3 · accepted · Accept

The authors have made the alterations suggested.